# Prolonged Survival Outcome in a Patient with Refractory Metastatic Colorectal Cancer Treated with Regorafenib Plus 5-Fluorouracil: A Case Report and Literature Review

**DOI:** 10.3390/reports8020059

**Published:** 2025-04-30

**Authors:** Abdullah Esmail, Bayan Khasawneh, Ebtesam Al-Najjar, Raed Zaidan, Maen Abdelrahim

**Affiliations:** 1Section of GI Oncology, Houston Methodist Neal Cancer Center, Houston Methodist Hospital, Houston, TX 77030, USA; aesmail@houstonmethodist.org (A.E.);; 2Department of Health, Human, and Biomedical Sciences, University of Houston–Clear Lake, Houston, TX 77058, USA

**Keywords:** colorectal cancer, metastasis, case report, regorafenib, 5-fluorouracil, 5-FU

## Abstract

**Background:** The use of regorafenib and 5-fluorouracil in the management of refractory metastatic colorectal cancer has gained increasing attention due to their demonstrated efficacy in extending the survival of patients with colorectal cancer. This study aims to discuss the effect of using regorafenib and 5-fluorouracil combination therapy in refractory metastatic colorectal cancer patients. **Case Presentation:** We present a case report of a 68-year-old female patient with KRAS G12D and PIK3CA mutations who was diagnosed with stage IV-C colon cancer. She was referred to hospice care and subsequently received therapeutic intervention with 56 cycles of regorafenib and 5-fluorouracil for 31 months while maintaining stable disease (SD). The patient exhibited good tolerance with minimal adverse effects, including Grade I-II Hand–Foot Syndrome. **Conclusions:** Our case showed the feasibility of using Regorafenib and 5-fluorouracil combination therapy in stage IV refractory metastatic colorectal cancer treatment, which resulted in an improvement in the overall survival after she was referred to Hospice care. Utilizing this case report may provide valuable input in managing refractory metastatic colorectal cancer, given the prolonged survival and the clinical meaningfulness of this regimen in our patient.

## 1. Introduction and Clinical Significance

Colorectal cancer (CRC) is the third most common cancer worldwide, and in the United States (US), it represents (7.8%) of all new cancer cases. CRC is the second leading cause of cancer death in the United States (US), and it poses a serious challenge for clinicians [1,2]. The median age at diagnosis for colorectal cancer patients is 66 years. About 20–40% of patients diagnosed with CRC eventually develop metastasis [3].

CRC is treated primarily with surgical removal of tumors. In the case of non-resectable CRC, standard therapies include chemotherapy, radiotherapy, and immunotherapy [4]. These therapies, however, have undesirable consequences since they are not specific to tumor cells only, thus causing damage to normal cells in the human body [5]. CRC therapies can be utilized in combinations to attain more desirable outcomes in treating cancer, but, unfortunately, more than half of patients develop multidrug-resistant CRC [6].

The mortality rate for CRC patients is still relatively high, despite significant advancements in CRC screening, surgical resection, and systemic therapies [7]. However, innovative therapies, such as targeted therapy and immune therapy, have transformed the landscape and improved patient outcomes. This concise introduction highlights the dynamic progress in CRC treatment, emphasizing a multidisciplinary approach and promising therapeutic advancements.

Multiple studies have assessed the efficacy of regorafenib, as monotherapy or in combination with other approved agents for patients with metastatic CRC (mCRC), by assessing the median Overall Survival (OS) in a number of patients who were followed for a 12-month period. The median OS improved in the treatment group as compared to the placebo group. In the CORRECT study, the median OS improved from 5.0 months in the placebo arm to 6.4 months in the treatment arm [8], while in the CONCUR study, the median OS improved from 6.3 months to 8.8 months [9]. Then came the SUNLIGHT study with a median OS improvement from 8.0 months to 10.0 months [10].

Regorafenib (Stivarga, BAY 73-4506) was approved by the Food and Drug Administration (FDA) in 2012 [11] as a follow-up treatment for mCRC in patients who did not respond to fluoropyrimidine, oxaliplatin, irinotecan-based chemotherapies, bevacizumab (anti-VEGF agent), or anti-EGFR therapy [12]. Regorafenib is a multi-kinase inhibitor that can block many kinases that are necessary for the growth of tumors [13,14]. This drug was approved after several trials where it demonstrated an increase in OS as compared to placebo [8,15]. Numerous research studies have endeavored to combine regorafenib with alternative medications, speculating that a potential synergistic impact of merging regorafenib with fluoropyrimidine-based chemotherapy regimens would yield superior results.

Treatment with 5-fluorouracil (5-FU), on the other hand, is known to improve survival in various cancers, including, but not limited to, colorectal, gastric, pancreatic, and breast cancers [16]. The largest impact of the drug has been reported in CRC [17]. 5-FU active metabolites disrupt both DNA and RNA synthesis through disruption of the folate metabolic pathway, resulting in the death of rapidly dividing cancer cells [18,19]. The overall response rate to 5-FU in advanced CRC is limited to 10–15%. Concomitant use of irinotecan and oxaliplatin in the 5-FU regimen was reported to improve survival among cancer patients. However, these combinations reported a higher incidence of adverse events [20,21]. The therapies are each approved individually to treat patients with mCRC, but their combined use has not yet been investigated [22]. In a study published by Cai et al., the researchers established 5-FU-resistant CRC cell lines HCT-116R and DLD-1R to assess the impact of the regorafenib effect. They found that regorafenib reduced stem-like characteristics, such as tumor sphere formation and the presence of side-population cells, in both cells lines. In vivo experiments demonstrated that combining regorafenib with 5-FU significantly inhibited tumor growth and decreased stemness markers in 5-FU-resistant DLD-1R cells. At the molecular level, regorafenib was linked to upregulation of the tumor suppressor and downregulation of WNT/β-catenin signaling. These results indicate that regorafenib, particularly when used with 5-FU, may effectively target cancer stem-like cells in CRC, providing a potential therapeutic strategy to overcome treatment resistance [23].

There are a few studies in the literature that discuss the efficacy of combining regorafenib and 5-FU in mCRC patients, the results of which provide a bright outlook on the effect of this combination regimen in achieving better outcomes, especially among patients with KRAS, BRAF, p53, and mismatch repair-deficient cells. However, such studies are limited [22].

In this case report, we present a unique case of a patient with refractory mCRC who was treated with a combination therapy of regorafenib plus intravenous 5-FU, which showed a significant survival outcome.

## 2. Case Presentation

This is a 68-year-old female patient who first presented in late 2012 with left flank pain and low-grade fevers, which had been in crescendo when she came into the emergency room. A Computed Tomography (CT) scan performed at that time showed a 13 cm pelvic mass (Figure 1) and thickening of the left colon (Figure 2) with associated lymphadenopathy. A subsequent colonoscopy was positive for well-differentiated invasive adenocarcinoma 50 cm (about 1.64 ft) from the anal verge. The mass was frond-like, partially obstructing, and circumferential. The patient underwent optimal resection with left hemicolectomy, along with Total Abdominal Hysterectomy and Bilateral Salpingo-oophorectomy (TAH and BSO), in late 2012. The pathology conducted after that was diagnostic of an infiltrating, well-differentiated mucinous adenocarcinoma of the colon measuring 10 cm, with bilateral ovarian metastases. The surgical margins were negative, and the tumor was staged as Stage IV-C. The tumor had positive perineural and lymphovascular invasion with fat infiltration. The patient had no change in bowel habits or rectal bleeding prior to surgery. There was no previous colonoscopy and negative personal or family history of cancer other than prostate cancer in her father. Molecular testing thereafter revealed mutated KRAS G12D, PIK3CA, Tumor Mutational Burden 8 (TMB8), MSI-stable, and was negative for NRAS and BRAF. The patient was then referred for adjuvant chemotherapy.

Adjuvant chemotherapy was initiated 4 weeks after surgery with 5-FU, leucovorin and oxaliplatin (FOLFOX), and bevacizumab. Adjuvant therapy was completed over 6 months, consisting of 12 cycles of FOLFOX combined with bevacizumab. Restaging Positron Emission Tomography and Computed Tomography (PET/CT) from the skull base to mid-thigh revealed no evidence of recurrent, residual, or metastatic disease.

One year later, the patient serologic and radiographic studies revealed evidence of disease progression, with a rise in the Carcinoembryonic Antigen (CEA) tumor marker, and imaging revealed metastatic disease in the liver, omentum, and mesentery. She received 7 months of 5-FU, leucovorin, irinotecan (FOLFIRI), and bevacizumab. FOLFIRI was discontinued due to poor tolerance and severe gastrointestinal side effects. Therapy with FOLFIRI and bevacizumab yielded an unknown response. She then was continued on single-agent bevacizumab for 6 months with persistent Stable Disease (SD).

After a 6-month chemotherapy break, radiographic imaging showed disease progression, with worsening omental nodularity and an increase in the size of anterior and left upper-abdominal lesions, which concluded disease progression. The patient commenced trifluridine/tipiracil therapy and has consistently received this treatment regimen since mid-2016, with bevacizumab for a brief period that was held at the end of the year 2016. She then received ramucirumab along with trifluridine/tipiracil in late 2017. Outside (PET/CT) showed SD-to-minimal progression, with a mild rise in CEA from 9 in 2016 to 37 in 2018. A few months later, the patient presented to another institution for a second opinion and was advised to continue treatment. Restaging conducted back then showed a continued response in the form of SD after being on trifluridine/tipiracil therapy for approximately 30 months.

The patient took ramucirumab monotherapy from late 2018 until early 2019, when trifluridine/tipiracil was resumed with ramucirumab. She continued to have SD and relatively good tolerance to treatment with an on-and-off approach, until she had chemical and radiological evidence of disease progression in 2019 when she was referred to Hospice care.

The patient decided to obtain another medical opinion on her case and was recommended for FOLFOX, irinotecan, and bevacizumab therapy, which were initiated in late 2019 until mid-2020. The Eastern Cooperative Oncology Group Performance Status (ECOG PS) was 0 at this point, with no major complaints. The patient had SD after 12 cycles of therapy. A few months later, she underwent a CT scan, which showed the progression of the disease.

Restaging CT Chest–Abdomen–Pelvis (CT CAP) in early 2021 showed disease progression and an increase in the size of multiple calcified abdominal masses (Figure 3). She redemonstrated moderate left hydroureteronephrosis, with a transition point in the left hemipelvis in the region of pelvic sidewall metastases. After progressing on the aforementioned regimens, she was transitioned to regorafenib 80 mg PO daily for days 1–7 of a 28-day cycle, then an escalated dose of 120 mg daily for 2 weeks and 1 week off, along with intravenous (IV) 5-FU 400 mg/m^2^ infusion starting in early 2021. The patient showed good tolerance for therapy and continued to have SD for approximately 12 months (Figure 4). The transition from disease progression to SD represented a favorable indicator of the drug’s efficacy in sustaining stability in this patient with mCRC. It was not until early 2022 when a CT scan was obtained, which showed mild progression in peritoneal metastasis, with persistent left-sided hydronephrosis due to a pelvic peritoneal mass causing obstruction. The scan also showed a circumferential peritoneal mass around the sigmoid colon, which might progress to obstruction. The patient thereafter continued to have SD, with no signs of bowel obstruction and stable severe left-sided hydronephrosis. A follow-up CT scan obtained in mid-2022 revealed extensive peritoneal disease and tumor implants, with involvement of the anterior abdominal wall, which had slightly increased in size with otherwise SD.

The patient continued to have SD for more than 12 months, until late 2023, when she presented to the ER complaining of abdominal pain, nausea, vomiting, and a decrease in bowel movements. Symptoms were similar to prior partial bowel obstructions. The patient was found to have a small and large bowel obstruction due to her malignancy. She did not recover bowel functions, despite conservative management, and eventually required NG tube placement to low intermittent suction. She was seen by interventional radiology and surgery. Both services were unable to place a venting G-tube due to peritoneal carcinomatosis. The patient was then started on Total Parenteral Nutrition (TPN) for nutritional support. A follow-up CT AP scan at that time revealed the mild progression of disease, with extensive widespread omental, peritoneal, retroperitoneal serosal, and subcapsular hepatic metastatic disease. CEA tumor marker levels were increased as well. These radiographic and chemical findings indicated disease progression and yielded discontinuation of regorafenib and 5-FU after receiving 56 cycles of therapy with persistent SD and good tolerance, with minimal adverse effects, including Grade I-II Hand–Foot Syndrome, which was managed by continuous hand cream application and topical steroids. The patient was switched to Pembrolizumab therapy, and after much discussion, she was designated DNR and was eventually placed into home Hospice in late 2023. The treatment timeline for the patient is indicated in Figure 5.

## 3. Discussion

This patient, diagnosed with stage IV CRC featuring metastasis, underwent multiple lines of treatment and eventually received regorafenib in combination with 5-FU therapy. This therapeutic regimen demonstrated a notable increase in OS, with the patient undergoing approximately 56 cycles of treatment and maintaining SD with minimal progression over a duration of about 31 months. These findings are promising and may reflect a tremendous improvement in enhancing survival outcomes among mCRC patients.

There are limited studies in the literature discussing the efficacy of a regorafenib plus 5-FU combination regimen in patients with mCRC. However, the available data suggest promising potential for improved outcomes with this regimen.

In a 2021 case study, a 64-year-old female patient with recurrent MSI-stable CRC with KRAS wild-type mutations, who had refractory CRC to oxaliplatin- and irinotecan-based chemotherapy combined with bevacizumab, demonstrated prolonged SD with third-line regorafenib and the PD-1 inhibitor sintilimab combination treatment [21]. Since MSI-stable tumors lack adequate immune system activation due to low mutation rates, these tumors are challenging to treat. In these settings, regorafenib is thought to confer sensitivity to tumors by its ability to reduce tumor-associated macrophages by blocking the colony-stimulating factor 1 (CSF-1) receptors. This case study concluded that the administration of regorafenib at an initial dose of 120 mg once daily, combined with sintilimab at 200 mg, resulted in CR conforming to the Response Evaluation Criteria in Solid Tumors (RECIST) version 1.1 criteria, and the patient exhibited no tumor progression [21].

Another study published in the year 2021 discussed a refractory mCRC patient who, after receiving first-line treatment with FOLFIRI and cetuximab for 12 cycles, subsequently received 4 cycles of FOLFOX and bevacizumab treatment [24]. These treatments, however, showed no response, and the patient continued to have disease progression. Subsequently, the patient received regorafenib as third-line therapy, initiating with 160 mg for two cycles and reducing the dose thereafter for a total of seventeen cycles. The patient achieved a CR and remained recurrence-free for 2 years post-cessation of treatment [24].

One more retrospective study published in 2016 enrolled 29 patients who had refractory mCRC in standard chemotherapies. The study had (66%) of patients receiving regorafenib at an initial reduced dose of 120 or 80 mg, while the other (34%) received 160 mg of regorafenib until they had disease progression or unacceptable toxic effects. The median duration of treatment was 2.5 months. Of those 29 patients, (55%) had the primary tumor in the colon, and the other (45%) had it in the rectum. Fifteen patients (51%) harbored RAS mutation. Treatment-related adverse events occurred in (86%) of patients, mostly fatigue, diarrhea, and Hand–Foot Syndrome. Regarding the efficacy of treatment, the best responses included SD in (24%) of patients, while (69%) of patients had progressive disease. No patient achieved partial or complete responses. The median OS was six months (95% CI, five-to-eight months) [25].

A study by Marks et al. presented the cases of two patients with mCRC who progressed on first- and second-line therapies with FOLFOX and FOLFIRI, respectively, and were consequently given cytotoxic and targeted therapies including regorafenib monotherapy, regorafenib with panitumumab, capecitabine plus dabrafenib, and trametinib. Both patients had progression following the administration of these therapies and were therefore offered regorafenib and 5-FU combination therapy. Subsequently, both patients had persistent SD with minimal toxicity. Notably, the study reported a synergistic effect with the combination of regorafenib and 5-FU in mCRC patients who had KRAS, BRAF, and p53 mutations, in addition to mismatch repair deficient cells. This highly correlates with our patient who had KRAS G12D mutation and achieved persistent SD response throughout her treatment with regorafenib plus 5-FU [22].

These studies align with our investigation into the effect of regorafenib, either as monotherapy or in combination with other systemic therapies, in enhancing survival outcomes for refractory mCRC patients. The dosage of regorafenib was judiciously adjusted to minimize the risk of potential adverse events.

## 4. Conclusions

Our case demonstrated that using regorafenib and 5-FU may significantly improve survival outcomes in patients with refractory mCRC. Notably, the prolonged survival observed in our patient is uncommon in patients with mCRC, where the outcomes are generally poor. Given the apparent efficacy seen in this case and limited prospective date available, we strongly advocate for continued exploration into the utilization of regorafenib and 5-fluorouracil for the treatment of refractory CRC. Further investigation through larger, controlled studies is essential to determine the therapeutic approach.

## Figures and Tables

**Figure 1 reports-08-00059-f001:**
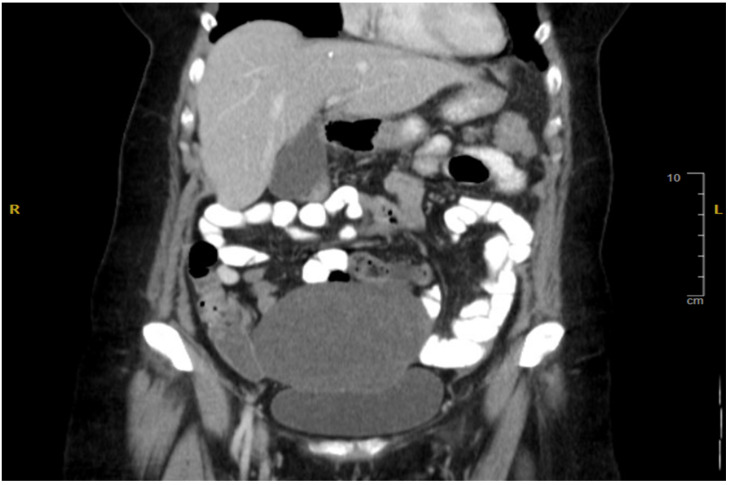
Coronal CT Abdomen Pelvis with contrast showing thickening of the left colon and pelvic mass.

**Figure 2 reports-08-00059-f002:**
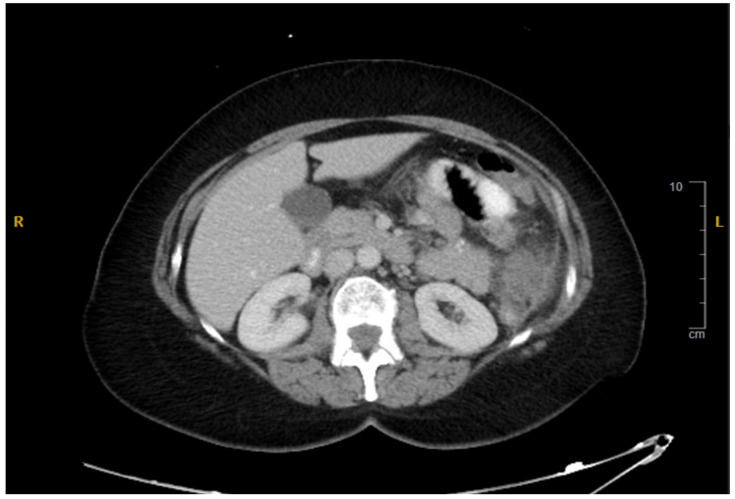
Axial CT Abdomen Pelvis with contrast indicating thickening of the left colon.

**Figure 3 reports-08-00059-f003:**
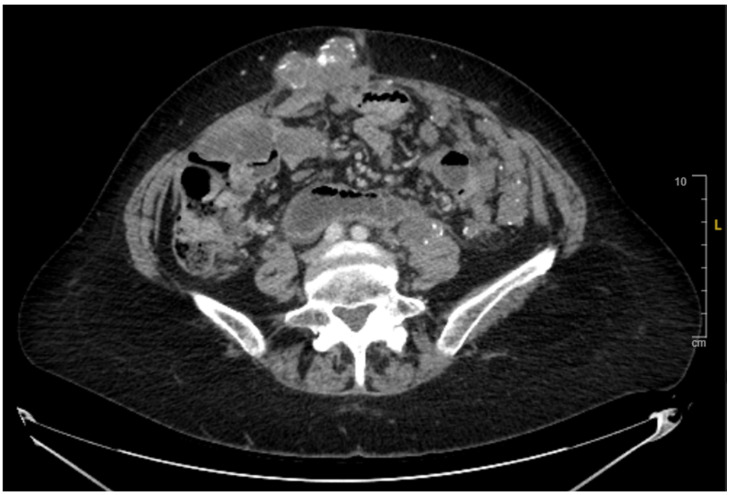
Axial CT Abdomen Pelvis indicating calcified abdominal masses.

**Figure 4 reports-08-00059-f004:**
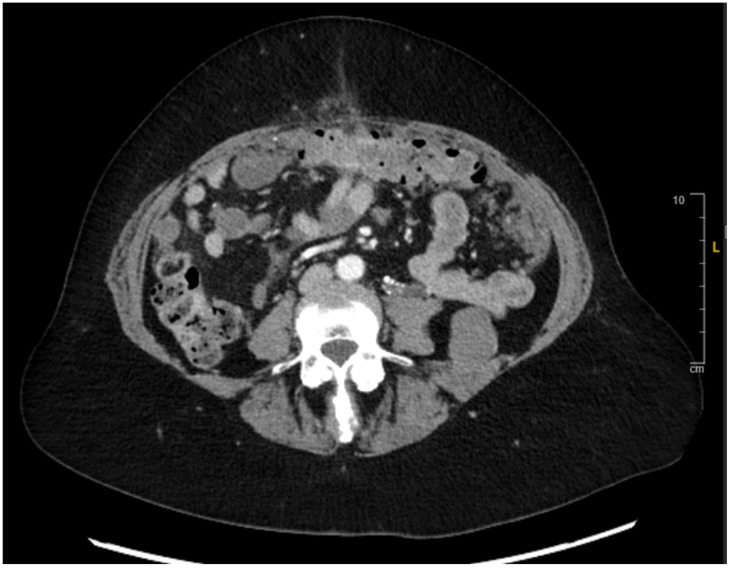
Axial CT Abdomen Pelvis indicating stable disease 1 year after treatment initiation.

**Figure 5 reports-08-00059-f005:**
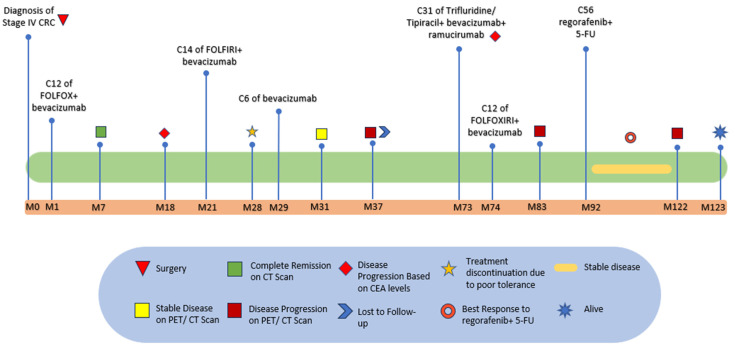
Treatment timeline detailing each line of therapy.

## Data Availability

The data of this case report that support the findings of “Significant Survival Outcome in Patient with Refractory Metastatic Colorectal Cancer Treated with Regorafenib Plus 5-Fluorouracil” are available upon request from the corresponding author, Maen Abdelrahim.

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
