# Peer review of "Prolonged Survival Outcome in a Patient with Refractory Metastatic Colorectal Cancer Treated with Regorafenib Plus 5-Fluorouracil: A Case Report and Literature Review"

_reports, 2025, doi:10.3390/reports8020059_

Round 1

Reviewer 1 Report

Comments and Suggestions for Authors

This is a very well-written case report. The overall length, the details of the case, the figure, and the discussion are all appropriate in terms of the detailing and the scientific soundness. It is an impressive case report that is presented well. I support the publication.

Author Response

Journal: Reports (ISSN 2571-841X)

Manuscript ID: reports-3576200

Type: Case Report

Title: Significant Survival Outcome in Patient with Refractory Metastatic Colorectal Cancer Treated with Regorafenib Plus 5-Fluorouracil: A Case Report and Literature Review

We sincerely thank all reviewers for their positive feedback on our study. We are especially grateful for the constructive comments that highlighted the strengths of our work and provided valuable suggestions to enhance its clarity and presentation. Your insights have significantly contributed to strengthening the final manuscript.

Review Report (Reviewer 1)

Comments and Suggestions for Authors

This is a very well-written case report. The overall length, the details of the case, the figure, and the discussion are all appropriate in terms of the detailing and the scientific soundness. It is an impressive case report that is presented well. I support the publication.

Answer: We deeply appreciate your valuable input and thank you for your thoughtful and encouraging feedback. We are truly grateful for your support and pleased to hear that you found the case report well-presented and scientifically sound.

Once again, we appreciate the reviewers' thoughtful comments and are confident that the revisions have improved the quality and impact of our manuscript.

Best regards,

The team

Reviewer 2 Report

Comments and Suggestions for Authors

Manuscript entitled “Significant Survival Outcome in Patient with Refractory Metastatic Colorectal Cancer Treated with Regorafenib Plus 5-Fluorouracil: A Case Report and Literature Review” by Esmail et al.

In this manuscript, the authors present the case of a patient with refractory metastatic colorectal cancer (mCRC) who achieved prolonged disease control with regorafenib and 5-fluorouracil combination therapy. While the topic is highly relevant and the paper addresses an important clinical challenge, the presentation has several notable weaknesses. Below is a structured critique.

Comments:

  1. Revise the Introduction to provide a more detailed explanation of the scientific rationale for combining a multi-kinase inhibitor (regorafenib) with a fluoropyrimidine (5-FU) in advanced mCRC, including any prior supporting evidence and discussion of known or proposed synergistic mechanisms that may enhance efficacy.
  2. Revise the Introduction to clearly identify the specific gap in current knowledge that this case aims to fill—explaining why this particular combination of regorafenib and 5-FU is noteworthy beyond the individual patient’s favorable response, especially in the context of existing treatments for refractory colorectal cancer.
  3. A major omission is the absence of a concise, tabulated summary of laboratory results. Since this is a case report focusing on treatment response and tolerability, it is essential to show time-trended values (e.g., CEA levels, hemoglobin, renal/liver function tests) to help readers track the clinical course.
  4. The patient’s journey—multiple lines of systemic therapy, numerous breaks, hospice referral, then additional therapy—could be clearer. The manuscript provides a timeline figure, but the explanation in the text is occasionally confusing. For instance, how exactly did the patient move from “hospice-eligible” to active multi-agent therapy again?
  5. While the authors note that regorafenib has anti-angiogenic and anti-proliferative properties, there is little exploration of why pairing it with 5-FU might outperform a single-agent approach. The potential synergy—especially in KRAS-mutant, PIK3CA-mutated backgrounds—needs more elaboration.
  6. The authors report only mild toxicity (e.g., Grade I/II Hand-Foot Syndrome) but do not discuss known concerns like hypertension, transaminitis, or fatigue, which often limit regorafenib dosing. Why was toxicity apparently so mild in this case? A deeper reflection would benefit clinicians.
  7. The manuscript’s conclusions about survival improvement come from a single case with no direct comparator. A formal limitations paragraph would underscore that these findings cannot be generalized without further study.
  8. There are scattered grammar inconsistencies (e.g., tense shifts, missing articles). A thorough copyedit would improve readability and ensure clarity.
  9. Compare this outcome more explicitly to prior combination studies or case reports.
  10. Conclusion: The statement about “significant survival outcome” might be strengthened by acknowledging how unusual or rare it is, especially in light of limited prospective data.

Author Response

Journal: Reports (ISSN 2571-841X)

Manuscript ID: reports-3576200

Type: Case Report

Title: Significant Survival Outcome in Patient with Refractory Metastatic Colorectal Cancer Treated with Regorafenib Plus 5-Fluorouracil: A Case Report and Literature Review

We sincerely thank all reviewers for their positive feedback on our study. We are especially grateful for the constructive comments that highlighted the strengths of our work and provided valuable suggestions to enhance its clarity and presentation. Your insights have significantly contributed to strengthening the final manuscript.

Review Report (Reviewer 2)

Comments and Suggestions for Authors

Manuscript entitled “Significant Survival Outcome in Patient with Refractory Metastatic Colorectal Cancer Treated with Regorafenib Plus 5-Fluorouracil: A Case Report and Literature Review” by Esmail et al.

In this manuscript, the authors present the case of a patient with refractory metastatic colorectal cancer (mCRC) who achieved prolonged disease control with regorafenib and 5-fluorouracil combination therapy. While the topic is highly relevant and the paper addresses an important clinical challenge, the presentation has several notable weaknesses. Below is a structured critique.

Comments:

  • Revise the Introduction to provide a more detailed explanation of the scientific rationale for combining a multi-kinase inhibitor (regorafenib) with a fluoropyrimidine (5-FU) in advanced mCRC, including any prior supporting evidence and discussion of known or proposed synergistic mechanisms that may enhance efficacy.

Response: We appreciated valuable comments. We have added a previous study to enhance our case.

  • Revise the Introduction to clearly identify the specific gap in current knowledge that this case aims to fill—explaining why this particular combination of regorafenib and 5-FU is noteworthy beyond the individual patient’s favorable response, especially in the context of existing treatments for refractory colorectal cancer.

Response: Thank you for your comments. We have added more details from the previous study to strength the scientific rationale of our case.

  • A major omission is the absence of a concise, tabulated summary of laboratory results. Since this is a case report focusing on treatment response and tolerability, it is essential to show time-trended values (e.g., CEA levels, hemoglobin, renal/liver function tests) to help readers track the clinical course.

Response: Thank you for the suggestion. Unfortunately, we do not have the complete set of laboratory results for this case, which is why a tabulated summary could not be provided. We have included all available data in the manuscript.

  • The patient’s journey—multiple lines of systemic therapy, numerous breaks, hospice referral, then additional therapy—could be clearer. The manuscript provides a timeline figure, but the explanation in the text is occasionally confusing. For instance, how exactly did the patient move from “hospice-eligible” to active multi-agent therapy again?

Response: Thank you for this insightful comment. After the patient was referred to hospice, she independently sought a second opinion, driven by a strong personal desire to pursue any available options to extend her survival. This led to the initiation of additional systemic therapy, despite her prior hospice eligibility.

  • While the authors note that regorafenib has anti-angiogenic and anti-proliferative properties, there is little exploration of whypairing it with 5-FU might outperform a single-agent approach. The potential synergy—especially in KRAS-mutant, PIK3CA-mutated backgrounds—needs more elaboration.

Response: We appreciate your comment. We have revised the manuscript to include brief discussion on the potential synergistic in patient in gene mutated.

  • The authors report only mild toxicity (e.g., Grade I/II Hand-Foot Syndrome) but do not discuss known concerns like hypertension, transaminitis, or fatigue, which often limit regorafenib dosing. Why was toxicity apparently so mild in this case? A deeper reflection would benefit clinicians.

Response: Thank you for this insightful comment. In this case, the only adverse effect reported was Grade I/II Hand-Foot Syndrome.

  • The manuscript’s conclusions about survival improvement come from a single case with no direct comparator. A formal limitations paragraph would underscore that these findings cannot be generalized without further study.

Response: Thank you for this comment. We have added a formal limitation in the conclusion section.

  • There are scattered grammar inconsistencies (e.g., tense shifts, missing articles). A thorough copyedit would improve readability and ensure clarity.

Response: Thank you for your valuable input. We have reviewed the manuscript and fixed all grammar errors.

  • Compare this outcome more explicitly to prior combination studies or case reports.

Response: We appreciate your valuable note. We have added a study in the discussion part to compare with our case.

  • Conclusion: The statement about “significant survival outcome” might be strengthened by acknowledging how unusual or rare it is, especially in light of limited prospective data.

Response: Thank you for the suggestion. We have revised the conclusion to highlight that prolonged survival is uncommon in refractory mCRC and to acknowledge the limited prospective data supporting this regimen.

Once again, we appreciate the reviewers' thoughtful comments and are confident that the revisions have improved the quality and impact of our manuscript.

Best regards,

The team

Reviewer 3 Report

Comments and Suggestions for Authors

This case report presents a compelling clinical scenario of a patient with refractory metastatic colorectal cancer (mCRC) who achieved a notable survival outcome following combination therapy with regorafenib and 5-fluorouracil (5-FU). The manuscript contributes to an important area of oncology and adds valuable insight to the limited literature on this therapeutic approach. However, there are several areas where the clarity, structure, and academic rigor of the manuscript could be enhanced to improve its scientific value and readability.

The title accurately reflects the content but could be streamlined for better impact. Consider: "Prolonged Survival in a Patient with Refractory Metastatic Colorectal Cancer Treated with Regorafenib and 5-Fluorouracil: A Case Report and Review"

In the abstract, the background is clear but could benefit from a more structured format (e.g., Background, Case Presentation, Conclusion). The phrase “tremendous impact” is subjective. A more scientific tone is recommended (e.g., “clinically meaningful survival extension”). Consider specifying the patient’s survival duration in the abstract for clarity.

The introduction provides a good overview of colorectal cancer epidemiology and current treatment challenges. A short paragraph summarizing the rationale for combining regorafenib and 5-FU specifically would strengthen the justification for the case study.

The case description is thorough and detailed. However, several aspects would benefit from editorial tightening:

  • Timeline: Although a figure is provided, including a brief table summarizing lines of treatment could enhance comprehension.
  • Language: There are multiple run-on sentences. For example, line 82-83 could be split and clarified.
  • Recommendation: Clarify and condense the narrative where possible. Avoid overly technical terms without definitions (e.g., “peritoneal carcinomatosis”) unless explained.
  • Figures: Ensure all figures have appropriate legends and high resolution.

The discussion appropriately contextualizes the case within existing literature. Strengthen the discussion by explicitly comparing the patient’s outcome (e.g., 31 months of stable disease) with outcomes from referenced studies. Expand slightly on the potential mechanisms of synergy between regorafenib and 5-FU, perhaps referencing recent preclinical findings.

The conclusion is sound, but somewhat repetitive. Focus on conciseness and avoid overstating findings from a single case report. Consider emphasizing the need for prospective evaluation or biomarker-driven studies.

Comments on the Quality of English Language

The manuscript would benefit significantly from a professional language edit. Common issues include:

Redundancies (e.g., “a case report detailing the management of a 68-year-old female patient...” could be simplified).

Informal language (“tremendous impact”) should be replaced with objective, scientific phrasing.

Typographical errors: “Flurouracil” should be corrected to “Fluorouracil” throughout.

Author Response

Journal: Reports (ISSN 2571-841X)

Manuscript ID: reports-3576200

Type: Case Report

Title: Significant Survival Outcome in Patient with Refractory Metastatic Colorectal Cancer Treated with Regorafenib Plus 5-Fluorouracil: A Case Report and Literature Review

We sincerely thank all reviewers for their positive feedback on our study. We are especially grateful for the constructive comments that highlighted the strengths of our work and provided valuable suggestions to enhance its clarity and presentation. Your insights have significantly contributed to strengthening the final manuscript.

Review Report (Reviewer 3)

Comments and Suggestions for Authors

This case report presents a compelling clinical scenario of a patient with refractory metastatic colorectal cancer (mCRC) who achieved a notable survival outcome following combination therapy with regorafenib and 5-fluorouracil (5-FU). The manuscript contributes to an important area of oncology and adds valuable insight to the limited literature on this therapeutic approach. However, there are several areas where the clarity, structure, and academic rigor of the manuscript could be enhanced to improve its scientific value and readability.

  • The title accurately reflects the content but could be streamlined for better impact. Consider: "Prolonged Survival in a Patient with Refractory Metastatic Colorectal Cancer Treated with Regorafenib and 5-Fluorouracil: A Case Report and Review"

Response: Thank you for your valuable comment. We have changed the title for a better impact.

  • In the abstract, the background is clear but could benefit from a more structured format (e.g., Background, Case Presentation, Conclusion). The phrase “tremendous impact”is subjective. A more scientific tone is recommended (e.g., “clinically meaningful survival extension”). Consider specifying the patient’s survival duration in the abstract for clarity.

Response: Thank you for your valuable comment. We have edited the abstract into a more structured format with a better scientific tone. We also added the survival duration for the patient per your suggestion.

  • The introduction provides a good overview of colorectal cancer epidemiology and current treatment challenges. A short paragraph summarizing the rationale for combining regorafenib and 5-FU specifically would strengthen the justification for the case study.

Response: Thank you for your comment. We have added a brief paragraph to the end of the introduction to clarify the rationale for combining regorafenib and 5-fluorouracil.

  • The case description is thorough and detailed. However, several aspects would benefit from editorial tightening:
  • Timeline: Although a figure is provided, including a brief table summarizing lines of treatment could enhance comprehension.
  • Language: There are multiple run-on sentences. For example, line 82-83 could be split and clarified.
  • Recommendation: Clarify and condense the narrative where possible. Avoid overly technical terms without definitions (e.g., “peritoneal carcinomatosis”) unless explained.
  • Figures: Ensure all figures have appropriate legends and high resolution.

  Response: Thank you for these valuable suggestions. We improved the language and the overall readability of the case report and ensured all the figures have appropriate legends and high resolution.

  • The discussion appropriately contextualizes the case within existing literature. Strengthen the discussion by explicitly comparing the patient’s outcome (e.g., 31 months of stable disease) with outcomes from referenced studies. Expand slightly on the potential mechanisms of synergy between regorafenib and 5-FU, perhaps referencing recent preclinical findings.

Response: We appreciate your comment. We have added additional case report studies in the discussion section to compare their findings with our case.

  • The conclusion is sound, but somewhat repetitive. Focus on conciseness and avoid overstating findings from a single case report. Consider emphasizing the need for prospective evaluation or biomarker-driven studies.

Response: We thank you for your comment. We have adjusted the conclusion and avoided overstating the findings.

Comments on the Quality of English Language

  • The manuscript would benefit significantly from a professional language edit. Common issues include:
  1. Redundancies (e.g., “a case report detailing the management of a 68-year-old female patient...” could be simplified).

Response: Thank you for your comment. We simplified it per your suggestion.

  1. Informal language (“tremendous impact”) should be replaced with objective, scientific phrasing.

Response: Thank you for your comment. We replaced it with more objective, scientific phrasing.

  1. Typographical errors: “Flurouracil” should be corrected to “Fluorouracil” throughout.

Response: Thank you for bringing this to our attention. We fixed it.

Once again, we appreciate the reviewers' thoughtful comments and are confident that the revisions have improved the quality and impact of our manuscript.

Best regards,

The team

Reviewer 4 Report

Comments and Suggestions for Authors

This manuscript presents a case report of a patient with refractory metastatic colorectal cancer who experienced an unusually prolonged survival after treatment with multiple lines of systemic therapy, including a combination of regorafenib and 5-FU.

The case is clinically intriguing, particularly given the extended survival duration and the use of regorafenib in conjunction with 5-FU. However, several critical areas require clarification and expansion to improve the report's scientific robustness and clinical relevance. Below are section-specific comments and suggestions:

Abstract

  • To enhance informativeness, consider including essential oncologic characteristics (genetic mutations), exact treatment duration, and precise clinical outcomes

Case Presentation

  • Is there available data on tumor marker levels, specifically CEA and CA 19-9, prior to surgery? Including these values would provide a clearer baseline for evaluating disease course and treatment response.
  • What was the rationale for including VEGF-targeted therapy (bevacizumab) in the adjuvant setting if the pelvic tumor mass was completely resected? This should be clarified.
  • Please indicate how many cycles of FOLFOX plus bevacizumab the patient received during the adjuvant treatment.
  • You state that the patient achieved “complete remission” following adjuvant therapy. However, such terminology is not typically applied in the adjuvant setting. Please revise this wording and clarify that imaging at follow-up showed no evidence of metastatic disease. Additionally, specify which imaging modality was used.
  • If baseline CEA values are available, please compare them with the values observed at the time of disease recurrence.
  • Given that more than a year had passed since adjuvant therapy, why was a reintroduction of the FOLFOX regimen not considered? Did the patient experience oxaliplatin-related toxicities, such as peripheral neuropathy?
  • How many cycles of FOLFIRI did the patient receive as first-line treatment for metastatic disease?
  • If the patient could not tolerate irinotecan, why was maintenance therapy with 5-FU and bevacizumab not continued?
  • Based on the imaging findings, which should be explicitly described, including the modality used, it appears the patient had peritoneal metastases. Was the patient evaluated by a surgical team for the possibility of undergoing HIPEC?
  • What was the reason for switching from trifluridine/tipiracil plus bevacizumab to ramucirumab?
  • You state: “The patient took ramucirumab monotherapy from late-2018 till early 2019 when trifluridine/tipiracil was resumed with ramucirumab.” If the patient had stable disease on ramucirumab monotherapy, what was the rationale for reintroducing trifluridine/tipiracil?
  • The sentence “The patient decided to get another medical opinion on her case” requires clarification. Was a rechallenge with previously administered therapies (FOLFOX, irinotecan, bevacizumab) undertaken? Please describe how this treatment was administered and how many cycles were given.
  • Regarding the statement “The patient was switched to Pembrolizumab therapy and after much discussion, she was made DNR,” please elaborate on the rationale for initiating immunotherapy with pembrolizumab in a patient with microsatellite-stable (MSS) disease, especially as a fifth-line treatment shortly before transitioning to best supportive care.

Discussion:

  • There is a lack of deeper discussion on the potential mechanisms of synergy between Regorafenib and 5-FU, which would enhance scientific interest and clinical applicability.
  • Some references mentioned in the discussion appear to have limited direct relevance to the combination regimen described. Including more targeted literature that is clearly related to the specific therapeutic approach tested would be beneficial

Author Response

Journal: Reports (ISSN 2571-841X)

Manuscript ID: reports-3576200

Type: Case Report

Title: Significant Survival Outcome in Patient with Refractory Metastatic Colorectal Cancer Treated with Regorafenib Plus 5-Fluorouracil: A Case Report and Literature Review

We sincerely thank all reviewers for their positive feedback on our study. We are especially grateful for the constructive comments that highlighted the strengths of our work and provided valuable suggestions to enhance its clarity and presentation. Your insights have significantly contributed to strengthening the final manuscript.

 Review Report (Reviewer 4)

Comments and Suggestions for Authors

This manuscript presents a case report of a patient with refractory metastatic colorectal cancer who experienced an unusually prolonged survival after treatment with multiple lines of systemic therapy, including a combination of regorafenib and 5-FU.

The case is clinically intriguing, particularly given the extended survival duration and the use of regorafenib in conjunction with 5-FU. However, several critical areas require clarification and expansion to improve the report's scientific robustness and clinical relevance. Below are section-specific comments and suggestions:

Abstract

  • To enhance informativeness, consider including essential oncologic characteristics (genetic mutations), exact treatment duration, and precise clinical outcomes

Answer: Thank you for your valuable input, we have included the oncologic characteristics, exact treatment duration, and the precise clinical outcomes for our patient.

Case Presentation

  • Is there available data on tumor marker levels, specifically CEA and CA 19-9, prior to surgery? Including these values would provide a clearer baseline for evaluating disease course and treatment response.

Answer: We appreciate your valuable note regarding the CEA and CA 19-9 levels prior to surgery. Unfortunately, these tumor maker levels are not available in the patient’s old medical records.

  • What was the rationale for including VEGF-targeted therapy (bevacizumab) in the adjuvant setting if the pelvic tumor mass was completely resected? This should be clarified.

Answer: Thank you for your valuable input. The patient was referred to us following his referral to hospice care thus, we do not have access to his old medical records other than the ones shared on his electronic medical records (EMRs).

  • Please indicate how many cycles of FOLFOX plus bevacizumab the patient received during the adjuvant treatment.

Answer: We appreciate your comment. The patient received 12 cycles of FOLFOX and bevacizumab. We added this information to the main text.

  • You state that the patient achieved “complete remission” following adjuvant therapy. However, such terminology is not typically applied in the adjuvant setting. Please revise this wording and clarify that imaging at follow-up showed no evidence of metastatic disease. Additionally, specify which imaging modality was used.

Answer: Thank you for your worthy comment. We corrected the terminology and added the type of imaging modality that was used in the main text per your valuable suggestion.

  • If baseline CEA values are available, please compare them with the values observed at the time of disease recurrence.

Answer: We appreciate your great suggestion. Unfortunately, the tumor maker levels are not available in the patient’s old medical records.

  • Given that more than a year had passed since adjuvant therapy, why was a reintroduction of the FOLFOX regimen not considered? Did the patient experience oxaliplatin-related toxicities, such as peripheral neuropathy?

Answer: Thank you for your valuable note. The patient came to us after he was referred to hospice care thus, we do not have access to his previous lab encounters other than the information that is already present in his EMRs. So, we cannot really report whether the patient experienced oxaliplatin-related neuropathy or not.

  • How many cycles of FOLFIRI did the patient receive as first-line treatment for metastatic disease?

Answer: We appreciate your comment. As the patient was not initially treated at our institution, we do not have an accurate estimation of how many cycles of FOLFIRI he received. However, based on the patient’s EMRs, he had received FOLFIRI for a total of 7 months with an unknown number of cycles.

  • If the patient could not tolerate irinotecan, why was maintenance therapy with 5-FU and bevacizumab not continued?

Answer: Thank you for your comment. The patient was not treated at our institution when he received FOLFIRI and bevacizumab so we cannot provide a justification to his prior treatment regimens.

  • Based on the imaging findings, which should be explicitly described, including the modality used, it appears the patient had peritoneal metastases. Was the patient evaluated by a surgical team for the possibility of undergoing HIPEC?

Answer: Thank you for this great question! However, the patient was referred to us in his late disease stages thus, we do not have enough information regarding the possibility of undergoing HIPEC. But to the best of our knowledge, she was seen by interventional radiology and surgery and both services were unable to place a venting G-tube due to the presence of peritoneal carcinomatosis, which may have impacted the decision to perform it.

  • What was the reason for switching from trifluridine/tipiracil plus bevacizumab to ramucirumab?

Answer: Thank you for your comment. The patient was not treated at our institution during his treatment with trifluridine/tipiracil plus bevacizumab or ramucirumab, so we cannot provide a reason to why he was switched from trifluridine/tipiracil plus bevacizumab to ramucirumab.

  • You state: “The patient took ramucirumab monotherapy from late-2018 till early 2019 when trifluridine/tipiracil was resumed with ramucirumab.” If the patient had stable disease on ramucirumab monotherapy, what was the rationale for reintroducing trifluridine/tipiracil?

Answer: Thank you for your comment. The patient did not receive his prior lines of treatment at our institution. However, based on his EMRs, the patient had trifluridine/tipiracil on hold while receiving ramucirumab per her oncologist’s advice to verify which medication is controlling the growth of the tumor.

  • The sentence “The patient decided to get another medical opinion on her case” requires clarification. Was a rechallenge with previously administered therapies (FOLFOX, irinotecan, bevacizumab) undertaken? Please describe how this treatment was administered and how many cycles were given.

Answer: Thank you for your comment. The patient was referred to us after she had progression of disease following 12 cycles of FOLFOX, irinotecan, and bevacizumab. The patient also preferred regorafenib and 5-FU based on her work conditions. Plus, the insurance did not approve bevacizumab. Therefore, she was started on regorafenib and 5-FU.

  • Regarding the statement “The patient was switched to Pembrolizumab therapy and after much discussion, she was made DNR,” please elaborate on the rationale for initiating immunotherapy with pembrolizumab in a patient with microsatellite-stable (MSS) disease, especially as a fifth-line treatment shortly before transitioning to best supportive care.

Answer: Thank you for your input. The patient was started on pembrolizumab after trying different lines of treatment. She received 1 cycle which was given based on the clinical judgment of the treating physician based on the possible microenvironmental changes that may have been present throughout her lines of treatment.

Discussion:

  • There is a lack of deeper discussion on the potential mechanisms of synergy between Regorafenib and 5-FU, which would enhance scientific interest and clinical applicability.

Answer: We appreciate your valuable input. We added more details to the discussion part to dive more deeply into the synergistic effect of regorafenib and 5-FU combination therapy based on your great comment.

  • Some references mentioned in the discussion appear to have limited direct relevance to the combination regimen described. Including more targeted literature that is clearly related to the specific therapeutic approach tested would be beneficial

Answer: We appreciate your valuable comment. We added new references to provide evidence and further support the matter discussed.

Once again, we appreciate the reviewers' thoughtful comments and are confident that the revisions have improved the quality and impact of our manuscript.

Best regards,

The team

Round 2

Reviewer 2 Report

Comments and Suggestions for Authors

The authors have adequately addressed my comments and concerns, the manuscript can be accepted for publication in present form. 

Reviewer 3 Report

Comments and Suggestions for Authors

The authors have made modifications according to the indications and recommendations of the reviewers, appropriately marked in the manuscript with colour coding. At present the manuscript can be considered for publication in this version.

Reviewer 4 Report

Comments and Suggestions for Authors

The additions and revisions in the new version of the manuscript have improved its clarity and scientific value. I'm pleased with the enhancements and believe that the authors have sufficiently addressed all concerns.

Cheers!